# Molecular Detection of *Cryptosporidium* spp. and Microsporidia in Human and Animal Stool Samples

**DOI:** 10.3390/microorganisms12050918

**Published:** 2024-04-30

**Authors:** María Teresa Gómez-Romano, Manuel Antonio Rodríguez-Iglesias, Fátima Galán-Sánchez

**Affiliations:** 1C.E.P. *Salus Infirmorum*, 11001 Cádiz, Spain; teresa.gomezromano@gmail.com; 2Servicio de Microbiología, Hospital Universitario Puerta del Mar, 11009 Cádiz, Spain; manuel.rodrigueziglesias@uca.es; 3Instituto de Investigación e Innovación en Ciencias Biomédicas de Cádiz (INiBICA), 11009 Cádiz, Spain; 4Facultad de Medicina, Universidad de Cádiz, 11003 Cádiz, Spain

**Keywords:** *Cryptosporidium* spp., *Enterocytozoon bieneusi*, farm animals, humans, pets, stool

## Abstract

*Cryptosporidium* spp. and Microsporidia are opportunistic microorganisms with remarkable zoonotic transmission potential due to their capacity to infect humans and animals. The aim of this study was to evaluate the prevalence of these microorganisms in stool samples of animal and human origin. In total, 369 stool samples (205 from human patients with diarrhea and 164 of animal origin) were included in the study. *Cryptosporidium* spp. and Microsporidia presence were determined by using multiplex nested PCR. Positive results were analyzed by using Sanger sequencing of the amplicon, utilizing BLASTN and ClustalX software to confirm identification. *Cryptosporidium* spp. were found in 0.97% and 4.26% of human and animal samples, respectively. *Enterocytozoon bieneusi* was detected in human and animal stools in 6.82% and 3.05% of the samples, respectively. No associations were found when analyzing the presence of *Cryptosporidium* spp. and *E. bieneusi* and the demographic and clinical variables of patients and animals. This study demonstrates the presence of these microorganisms in human and animal samples from different species, and the most interesting findings are the detection of *Cryptosporidium* spp. in pets (e.g., rodents) that are not usually included in this type of study, and the identification of *E. bieneusi* in patients with diarrhea without underlying disease.

## 1. Introduction

*Cryptosporidium* spp. are protozoan parasites within the phylum *Apicomplexa* and they infect the microvilli of the gastrointestinal epithelium of a wide range of vertebrates, including humans [1]. Profuse diarrhea has become the most common symptom. Moreover, extraintestinal cryptosporidiosis have been reported, with lungs, the hepatobiliary system, and pancreas mainly being affected [2]. Cryptosporidiosis have been described in immunocompromised as well as in immunocompetent people [3,4], and are associated with several different types of exposures. Large outbreaks of cryptosporidiosis have been associated with person-to-person transmission and exposure to recreational waters, animal and environmental sources, and food [4,5,6,7]. While cryptosporidiosis is recognized as being among the most common causes of human parasitic diarrhea in the world, there is currently limited knowledge on *Cryptosporidium* infection mechanisms, a scarcity of diagnostic methods, and a need for additional therapeutic options [8].

Microsporidia are a ubiquitous group of spore-forming obligate intracellular microorganisms that infect a wide range of vertebrates and invertebrates. They belong to the phylum *Microsporidia* within the kingdom Fungi [9]. However, at the taxonomic level, there are some contradictions that remain in doubt. More than 220 genera, which include an additional 1700 species, have been described to date. Nine of these genera (*Encephalitozoon*, *Enterocytozoon*, *Pleistophora*, *Trachipleistophora*, *Vittaforma*, *Anncaliia*, *Endoreticulatus*, *Tubulinosema, and Microsporidium)* [10] and seventeen species are related to human infection, with *Enterocytozoon bieneusi* and *Enterocytozoon intestinalis* being the most common [11]. These species are responsible for an extensive variety of systemic and non-systemic diseases. Although chronic diarrhea is the clinical manifestation that best correlates with infection due to these microorganisms, ocular, breathing, renal, and nervous system infections have also been described [2,5]. Since the discovery of the species *E. bieneusi*, the number of patients affected by this microorganism has increased considerably. As a result, Microsporidia have become causal agents of intestinal and systemic infections in patients with human immunodeficiency virus (HIV) and other immunocompromised individuals. Additionally, the presence of Microsporidia has been described in the feces of immunocompetent people [12]. Transmission occurs mainly through fecal–oral routes; accordingly, sources of infection are usually infected people or animals. Moreover, even food or water contaminated with spores could be an infection focus [13]. Several works have shown the zoonotic potential of Microsporidia [13,14].

Although *Cryptosporidium* spp. and Microsporidia are opportunistic microorganisms with a potential for zoonotic transmission, information on their prevalence in pets and other animals in close contact with humans and their contribution as an etiological agent of diarrhea in humans (mainly due to the lack of routine diagnosis in clinical laboratories) is scarce. In Spain, there has been a remarkable increase in the number of pet owners in recent years (https://www.anfaac.org/datos-sectoriales/ (accessed on 1 February 2024). In this study, we evaluate the prevalence of these microorganisms in stool samples from animal and human origin.

## 2. Materials and Methods

### 2.1. Origin of Samples

Feces of animal and human origin were collected over three years (2017, 2018, and 2022). The samples of animal origin refer to pets and farm animals collected in the province of Cádiz, southern Spain, whose owners were pupils of educational institutions. Dog, cat, cow, pig, and goat samples were collected by swab directly from feces, and chicken samples were recollected from cottages, in a collective way; dove feces were sampled directly from deposition places.

The human samples belong to surplus diarrhea samples. These samples were sent in Cary Blair transport medium to the Clinical Microbiology Laboratory of the Hospital Universitario Puerta del Mar for routine stool culture (for the diagnosis of *Salmonella enterica*, *Shigella* spp., *Campylobacter* spp., *Aeromonas* spp., *Vibrio* spp., and *Yersinia* spp.) and included all the samples received in the laboratory during one day of each month. All the surplus diarrhea and animal samples included in the study were immediately processed for DNA extraction.

### 2.2. Sample Size

Using the G-Power 3.1 program, a minimum sample size of n = 112 was calculated for an effect size of 0.3, alpha = 0.05, and a power of 0.90. Nevertheless, an attempt was made to include as many patients and animals as possible, so the final samples consisted of a total of 205 human and 164 animal samples.

### 2.3. Statistical Analysis

The association between the presence of *Cryptosporidium* and Microsporidia and demographic and clinical variables were evaluated by means of χ2 tests of homogeneity or Fisher’s exact test when appropriate. All the statistical analyses were performed using IBM SPSS software v. 26 (IBM Corp., Armonk, NY, USA). In all cases, *p* values ≤ 0.05 were considered significant.

### 2.4. DNA Extraction

For DNA extraction, ASL buffer (QIAGEN GmbH, Hilden, Germany) was used, following the manufacturer’s instructions. Briefly, 500 µL of ASL buffer was added to 15 µL of the stool sample in Cary Blair medium (human samples) or a feces suspension in sterile distilled water (animal samples); the mix was vortexed for 20 s, incubated for 10 min at 95 °C, and then centrifuged at 13,000 for 5 min. The supernatant was used for DNA extraction using the EZ1 automated extraction system (QIAGEN).

### 2.5. Bacterial 16S rRNA Gene PCR

In order to discard PCR inhibitors, a 16S rRNA gene PCR was performed on all the samples. The PCR reactions were set in a 25 µL final volume. Each PCR tube contained 2.5 µL of My Taq HS DNA polymerase (Bioline Meridian Bioscience, Cincinnati, OH, USA), 5 µL of DNA, 5 µL of 5× PCR buffer, 0.25 µM of primers fD1 (AGAGTTTGATCCTGGCTCAG), and rD2 (ACGGCTACCTTGTTAGCACTT). A total of 40 cycles were carried out, each consisting of 94 °C for 15 s, 60 °C for 15 s, and 72 °C for 30 s, with an initial hot start at 94 °C for 5 min and a final extension at 72 °C for 7 min. PCR amplification products were resolved by electrophoresis using a 1% agarose gel.

### 2.6. Multiplex Nested PCR

A multiplex nested PCR was performed for the detection of *E. bieneusi*, *Encephalocytozoon,* and *Cryptosporidium* spp. as proposed by Rubio et al. [5]. In each run, positive (kindly provided by Dr. David Carmena, from the Parasitology Reference and Research Laboratory, National Center for Microbiology, Majadahonda, Madrid, Spain) and negative controls were included. The PCR reactions were set in a 25 µL final volume. Each PCR tube contained 5 µL of DNA, 0.5 µL of My Taq HS DNA polymerase (Bioline Meridian Bioscience), 5 µL of 5xMy Taq PCR buffer, 1 µM of primer Univ1 (GATTCCGGAGAGGGAGCC), and 0.5 µM of primers Crypto1 (CAAGGCATATGCCTGCTTTAAGC), Entero1 (CGCCAGTCTATACTACACTCCCTATCC) and Encef1 (CTTCGCTTCTSTYCGTCCAG). A total of 40 cycles were carried out, each consisting of 94 °C for 30 s, 61 °C for 30 s, and 72 °C for 30 s, with an initial hot start at 94 °C for 7 min and a final extension at 72 °C for 10 min. The second PCR reaction conducted was the identification reaction and it incorporated the products of the first reaction (2 µL) along with 0.5 µL of My Taq HS DNA polymerase (Bioline Meridian Bioscience), 5 µL 5× MyTaq PCR buffer, and 0.75 uM of primer Crypto2 (CCTCCAATTGATACTTGTAAAGGG), 0.5 µM of primers Entero2 (GGTCTTTACACTCAGGTCTTATACTCA) and Encef2(CTTCRTATTTCACCCCTCGC), plus 1 µM of primer Univ2 (CGGAGAGGGAGCCTGAG). A total of 35 cycles were carried out, each consisting of 94 °C for 15 s, 58 °C for 15 s, and 72 °C for 20 s, with an initial hot start at 94 °C for 7 min and a final extension at 72 °C for 10 min. The PCR amplification products were resolved by using electrophoresis with a 2% agarose gel. Positive results were confirmed by using Sanger sequencing of the amplicon, utilizing BLASTN (blast.ncbi.nlm.nih.gov) and ClustalX (www.clustal.org) software to confirm identification and avoid false positive results due to non-specific reactions.

## 3. Results

Two hundred and five samples of human origin were analyzed. Epidemiological characteristics, such as the age and hospitalization of the patients included in the study, are presented in Table 1. In detail, 48% of the samples came from women, while samples from men accounted for 51.7% of the total. In addition, 5.8% of the samples came from oncologic patients, 3.9% from patients affected by chronic kidney diseases, 0.97% from patients diagnosed with chronic hepatitis, and 0.97% from patients diagnosed with HIV.

One hundred and sixty-four samples from different species of animal origin were collected and analyzed (Table 2). In detail, 50% of these samples came from dogs, which were the most representative species, followed by birds and cats, which represented 16.5% and 14% of the samples, respectively. Samples from rodents represented 5.5% of the total, and the remaining (14%) were from different farm animals.

### 3.1. Cryptosporidium *spp.* and Microsporidia Detection in Human Fecal Samples

*Cryptosporidium* spp. were detected in 2 samples (0.97%, 95% CI 0.27–3.49) and *E. bieneusi* was detected in 14 samples (6.82%, 95% CI 4.11–11.14) of the 205 human feces samples. The epidemiological and clinical characteristics of the patients with positive results are shown in Table 3.

The χ2 tests did not show any relationship (*p* > 0.05) between the frequency of *E. bieneusi* and the underlying disease, sex, age, or origin of the patients.

Fisher’s exact test did not show any relationship (*p* > 0.05) between the frequency of *Cryptosporidium* spp. and the underlying disease, sex, age, or origin of the patients.

### 3.2. Cryptosporidium *spp.* and Microsporidia Detection in Animal Fecal Samples

*Cryptosporidium* spp. and *E. bieneusi* were detected in 4.26% (95% CI 2.08–8.55) and 3.04% (95% CI 1.31–6.94) of the samples of animal origin, respectively. The epidemiological and clinical characteristics of the animals with positive results are presented in Table 4.

The χ2 tests did not show any relationship (*p* > 0.05) between the frequency of *E. bieneusi* and the sex, age, or type of population of the animals.

Fisher´s exact test did not show any relationship (*p* > 0.05) between the frequency of *Cryptosporidium* spp. and the sex, age, or type of population of the animals.

## 4. Discussion

In Spain, there was a remarkable increase in the number of pet owners. In 2021, the numbers of dogs and cats were estimated to be approximately 9.3 and 5.8 million, respectively (https://www.anfaac.org/datos-sectoriales/ accessed on 1st February 2024.

Despite companionship from these animals, which provide a clear benefit to people, pets can carry diseases that spread between them and humans, presenting a potential threat to public health [15].

Our study has demonstrated the presence of *Cryptosporidium* spp. and *E. bieneusi* in humans and domestic and farm animals. These animals are in close contact with human populations and could be an important zoonotic factor for infection and a possible source of environmental contamination [16].

In our study, no relationships between the different demographic and clinical variables of the human and animal populations with *Cryptosporidium* spp. were found. We detected *Cryptosporidium* spp. in only two human samples from adult oncologic patients. However, no positive samples from pediatric feces were detected, in accordance with a previous study performed in the community of Madrid, where the presence of *Cryptosporidium* spp. in children from 1 to 16 years old was very low (0.9%) (13/1512) [17]. In a study carried out in Brazil that included children with underlying diseases, such as diarrhea, cancer, HIV, and malnutrition, the prevalence was 2.2% (14/626) in this age range [18]. In samples of animal origin, we detected the presence of *Cryptosporidium* spp. in a cat suffering from diarrhea (4.34%; 1/23) as well as in three dogs (3.61%; 3/83). A recent study carried out in Poland showed similar results, describing the presence of this microorganism in two felines, also affected with diarrhea, (n = 101; 2%), and nine asymptomatic dogs (n = 264; 3.4%) [19]. This study emphasizes the presence of diarrhea in both felines, just as described in our study. Moreover, another study performed in Austria showed a prevalence of 1.7% (37/298) when cats were analyzed [20]. A study carried out in 2010 highlighted the presence of these parasites (9%) in fecal samples from dogs and cats taken from public parks in Madrid [14].

In addition, *Cryptosporidium* spp. was detected in two farm animals: a lactating pig (20%; 1/5) and a goat (16.66%; 1/6). A similar study carried out on farms in China found a prevalence of 2.7% (8/299) in lactating pigs [21]. The prevalence of this microorganism in goats is high, as was revealed in a study carried out in Ecuador where 10.49% of the goats were infected by *Cryptosporidium* spp. [22]. A similar study described a prevalence of 14.3% (1/7) in different farms located in northern Spain (Galicia) [23].

In our study, the presence of *Cryptosporidium* spp. was described in a dwarf winter white Russian hamster for the first time in Spain, which was the only one that was included. A recent study in China that included these rodents [24] described an incidence of 39.32% in this species (138/351). However, there is a lack of information focused on this kind of rodent, and it would be interesting to include rodents in future studies due to their possible implications for human health, as they are usually found as pets.

A major limitation of our study is that no species identification in *Cryptosporidium* was done, so it is unclear which relevance the observation of *Cryptosporidium* spp. in animals has.

In our study, when infections by *E. bieneusi* were analyzed, the results revealed interesting data, although no relationship between the different demographic and clinical variables of the human and animal population with *E.bieneusi* was found. We have found that the percentage of infected patients aged 14 years and older was lower than that in pediatric patients (5.8% vs. 11.7%) and included two adult oncologic patients (2/171). One study carried out in Egypt described a prevalence of 4.6% (27/585) in pediatric patients and noted that children from rural zones, who are usually more in contact with animals than children from urban areas, have a higher infection rate [25]. This fact is corroborated by a study carried out in China, where a higher infection rate by *E. bieneusi* in rural compared to urban areas was estimated [26]. Our study included children from urban areas of two cities with more than 95,000 citizens.

It has been shown that the Microsporidia species constitute one of the more important opportunistic pathogens due to their ability to cause grave disease after infection, especially in immunocompromised patients [27]. Therefore, the clinical course of microsporidiosis depends on the host immune state, where the infections occur, and the Microsporidium species implicated [28]. The number of patients not infected by HIV who present other immunodeficiency problems, such as transplanted or oncologic patients, has also increased [5].

A recent study carried out in Spain showed the relationship between colon cancer and Microsporidia infection, which was revealed through an analysis of tissue samples [29]. The presence of healthy carriers has also been documented, which clearly shows the presence of Microsporidia in the feces of immunocompetent patients and that microsporidiosis may not be linked to any clinical symptoms in healthy populations, as it has been detected in individuals infected with *E. bieneusi* who are totally asymptomatic [12].

Remarkably, in our study, the majority of patients with *E. bieneusi* did not have any serious underlying disease, and other diarrhea-causing enteropathogens were not found. However, more studies are necessary to clarify the need to include the determination of this microorganism in the routine study of diarrhea. It could be especially interesting when no other enteropathogens are found in the usual determinations.

Regarding the prevalence of *E. bieneusi* in different animals, our results reveal its presence to be mainly in dogs (4.81%; 4/83). Several studies carried out in Spain show that the prevalence of this microorganism in dogs varies considerably, with percentages between 0.8% and 11.7% [11,15,23]. These studies have included pets as well as animals from the street, with the last group being the largest carrier of the microorganism. In China, other studies have revealed an incidence of 6.74% (18/2067), and most infections occurred in males younger than one year old that were collected from a pet market [30]. Another study carried out in Australia showed an incidence of 4.4% (15/342) [31]. However, in a recent study carried out in Portugal, no positive samples for *E. bieneusi* were found in 46 fecal samples [32]. Even so, our study indicates that dogs in urban settings are also prone to harbor *E. bieneusi* infections in addition to stray dogs, which may be more affected due to greater exposure to the parasite and poorer standards of care [15]. In addition, in our study, we detected the presence of *E. bieneusi* in the feces of a pig (20%; 1/5). In comparison to these results, a previous study carried out in southern Spain (Cordoba) revealed an incidence of 22.6% in these animals [33].

On the other hand, due to the limited number of swine feces samples included in this study, no significant relationships between prevalence and public health were detected. Therefore, more samples are needed in future studies to provide more information about *E. bieneusi* infection in pigs as well as in other animals and humans.

## 5. Conclusions

This study evidences the presence of *Cryptosporidium* spp. and *E. bieneusi* in human and animal samples. On the one hand, the detection of *Cryptosporidium* spp. in pets that are not usually included in this type of study, such as rodents, is one of the most interesting findings, and, on the other hand, the identification of *E. bieneusi* in patients with diarrhea of different age groups and without underlying disease is a highly interesting finding also. It would be desirable to carry out further studies on these two aspects, analyzing the risk of zoonotic transmission of certain types of pets and the convenience of including the determination of *E. bieneusi* in the routine study of parasites since it may represent a problem of public health concern.

## Figures and Tables

**Table 1 microorganisms-12-00918-t001:** Number of hospitalized and non-hospitalized patients by age range.

Age Range	Outpatient (n; %)	Inpatient (n; %)	Total
0–14	32; 19.4	2; 5	34
15–40	44; 26.6	2; 5	46
41–65	59; 35.7	14; 3	73
>66	30; 18.2	22; 5	52
Total	165	40	205

**Table 2 microorganisms-12-00918-t002:** Animals populations investigated in the present study.

	Dogs (n = 83)	Cats (n = 23)	Rodents ^a^ (n = 3)	Lagomorphs ^b^ (n = 6)	Birds ^c^ (n = 27)	Others ^d^ (n = 22)
Pets	100%	100%	100%	100%	22%	
Farm					78%	100%

^a^ 2 hamsters and 1 guinea pig, ^b^ 6 rabbits, ^c^ 1 turtledove, 2 ducks, 1 turkey, 1 canary, 2 lovebirds, 1 parrot, 1 kestrel, 4 pigeons, and 14 group samples of chicken coops; ^d^ 5 pigs, 6 goats, 11 cows.

**Table 3 microorganisms-12-00918-t003:** Positive samples for *Enterocytozoon bieneusi* and *Cryptosporidium* spp. in fecal samples from humans.

ID	Sex	Age	Underlying Disease	Origin	Microorganism
5	Female	61 years	Lymphoma	Inpatient	*Cryptosporidium* spp.
20	Male	22 years		Outpatient	*E. bieneusi*
26	Male	80 years		Inpatient	*E. bieneusi*
42	Female	24 years		Outpatient	*E. bieneusi*
101	Male	1 month		Outpatient	*E. bieneusi*
106	Female	57 years		Outpatient	*E. bieneusi*
111	Female	37 years		Outpatient	*E. bieneusi*
115	Female	21 years		Outpatient	*E. bieneusi*
118	Male	10 years		Outpatient	*E. bieneusi*
120	Female	7 years		Outpatient	*E. bieneusi*
125	Female	32 years		Outpatient	*E. bieneusi*
128	Male	89 years	Melanoma	Outpatient	*E. bieneusi*
129	Male	63 years	Lung adenocarcinoma	Outpatient	*E. bieneusi*
131	Female	42 years		Outpatient	*E. bieneusi*
180	Male	6 years		Outpatient	*E. bieneusi*
186	Male	59 years	Metastatic Colon Adenocarcinoma	Inpatient	*Cryptosporidium* spp.

**Table 4 microorganisms-12-00918-t004:** Positive samples for *Enterocytozoon bieneusi* and *Cryptosporidium* spp. in fecal samples from animals.

ID	Host	Breed	Age	Microorganism	Diarrhea	Type of Population
17	Cat	Common European	3 months	*Cryptosporidium* spp.	yes	Pet
26	Dog	English setter	10 years	*E. bieneusi*	no	Pet
29	Dog	Yorkshire	16 months	*Cryptosporidium* spp.	no	Pet
67	Pig	Iberian Pork	Breeding	*Cryptosporidium* spp.	unknown	Farm
68	Pig	Iberian Pork	Breeding	*E. bieneusi*	unknown	Farm
69	Goat	Unknown	Breeding	*Cryptosporidium* spp.	unknown	Farm
70	Dog	Hungarian Shorthaired Pointer	1 year	*Cryptosporidium* spp.	no	Pet
75	Dog	Mongrel	4 years	*E. bieneusi*	no	Pet
94	Dog	Golden Retriever	14 months	*E. bieneusi*	no	Pet
96	Dog	Ratonero Andaluz	15 years	*Cryptosporidium* spp.	no	Pet
123	Hamster	Darwf winter White Russian	1 year	*Cryptosporidium* spp.	no	Pet
135	Dog	Mongrel	2 years	*E. bieneusi*	no	Pet

## Data Availability

Data are contained within the article.

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
