# Peer review of "Molecular Detection of Cryptosporidium spp. and Microsporidia in Human and Animal Stool Samples"

_microorganisms, 2024, doi:10.3390/microorganisms12050918_

Round 1
Reviewer 1 Report
Comments and Suggestions for Authors
First impression of reading present is just the "Similar But Different".
The genus Cryptosporidium belongs to one of the non-typical apicomplexian taxa among the kingdom Protista.
On the other hand, the Microsporidia is regarded as a strange taxon of the kingdom Fungi.
Even so both are alike in appearance and in tiny size, but, they are absolutely different in nature.
Hence, the authors have to rewrite two individual short note (both are NOT original article, because of the results themselves are not so great),
namely one for protist genus Cryptosporidium, another for the fungal genus Enterocytozoon.
Add to this, the lines of 39 & 40, ...phylum Microsporidia within the kingdom Fungi. However... The word Fungi was in Italic type, but that
in specific name, so the do not use such letters here.
Author Response
Dear editor and reviewers, we are very grateful for your recommendations, and we strongly believe they are relevant. We have seriously considered all your points. Please find the detailed responses below and the corresponding revisions/corrections highlighted/in track changes in the re-submitted files.
Comments 1: First impression of reading present is just the "Similar But Different". The genus Cryptosporidium belongs to one of the non-typical apicomplexian taxa among the kingdom Protista. On the other hand, the Microsporidia is regarded as a strange taxon of the kingdom Fungi. Even so both are alike in appearance and in tiny size, but, they are absolutely different in nature. Hence, the authors have to rewrite two individual short note (both are NOT original article, because of the results themselves are not so great), namely one for protist genus Cryptosporidium, another for the fungal genus Enterocytozoon.
Response 1: Thank you for your comment. As you indicated, these are two microorganisms of different nature, but we believe that they share some epidemiological and clinical similarities that justify their joint study, such as the type of host, their association with diarrhea, their zoonotic potential and their ability to produce serious infections in immunosuppressed patients. For this reason, we thought it would be interesting to describe the results together, since both microorganisms are also agents not generally included in routine studies of diarrhea.
Comments 2: Add to this, the lines of 39 & 40, ...phylum Microsporidia within the kingdom Fungi. However... The word Fungi was in Italic type, but that in specific name, so the do not use such letters here.
Response 2: The text has been corrected as suggested (line 44)
Reviewer 2 Report
Comments and Suggestions for Authors
The article “Molecular Detection of Cryptosporidium spp. and Microsporidia in Human and Animal Stool Samples” aims to determine, by multiplex nested PCR and DNA sequencing, the prevalence of the agents, Cryptosporidium spp. and Microsporidia in 205 human stool samples and 154 animal stool samples. The results are interesting, but some statistical analysis should be incorporated to make the results less descriptive and analyze the association between variables. I recommend improving the result section before it is published.
Major comments:
- In the M&M section, the authors must explain how the sample size in animals and humans was calculated.
- Line 91-92, please indicate the information about the primers used to amplify Cryptosporidium and Microsporidia in multiplex nested PCR. Additionally, indicate the conditions used in the PCR in detail.
- In the results section, authors indicate some epidemiological data about the patients, such as sex, age, underlying disease, origin, and microorganism detected. Highlights the presence of Cryptosporidium only in patients with underlying disease. It could be interesting to perform a statistical analysis to elucidate if the underlying disease is a determinant of finding Cryptosporidium or if there is some association.
- The same comment applies to samples from animals for which more variables were registered (host, breed, age, microorganism, diarrhea, type of population) for both pathogens.
- The sequences of DNA were analyzed by Sanger, but the Cryptosporidium species are not indicated in the result section. Please explain why or clarify in M&M section if all positive samples for both pathogens were sequenced.
- Ethical and bioethical certifications were not included in the article. Please add this information.
Minor suggestions:
- Line 30, please replace “Cryptoporidiosis” by “Cryptosporidiosis”.
- Line 40, please add a reference after “kingdom Fungi”.
- Line 45, write genera and species the first time they are mentioned (E. bieneusi and E. intestinalis).
- Line 85, please correct “1,3000”.
Author Response
Dear editor and reviewers, we are very grateful for your recommendations, and we strongly believe they are relevant. We have seriously considered all your points. Please find the detailed responses below and the corresponding revisions/corrections highlighted/in track changes in the re-submitted files.
Comments 1: The article “Molecular Detection of Cryptosporidium spp. and Microsporidia in Human and Animal Stool Samples” aims to determine, by multiplex nested PCR and DNA sequencing, the prevalence of the agents, Cryptosporidium spp. and Microsporidia in 205 human stool samples and 154 animal stool samples. The results are interesting, but some statistical analysis should be incorporated to make the results less descriptive and analyze the association between variables. I recommend improving the result section before it is published.
Response 1: The section “2.3 Statistical Analysis” has been added, as suggested (line 92). The results are described in lines 173 and 187.
Major comments:
Comments 2: In the M&M section, the authors must explain how the sample size in animals and humans was calculated.
Response 2: The section “2.2 Sample Size” has been added, as suggested (line 85)
Comments 3: Line 91-92, please indicate the information about the primers used to amplify Cryptosporidium and Microsporidia in multiplex nested PCR. Additionally, indicate the conditions used in the PCR in detail.
Response 3: Information about primers and conditions used in the PCR have been added to the text (line 122)
Comments 4: In the results section, authors indicate some epidemiological data about the patients, such as sex, age, underlying disease, origin, and microorganism detected. Highlights the presence of Cryptosporidium only in patients with underlying disease. It could be interesting to perform a statistical analysis to elucidate if the underlying disease is a determinant of finding Cryptosporidium or if there is some association. The same comment applies to samples from animals for which more variables were registered (host, breed, age, microorganism, diarrhea, type of population) for both pathogens.
Response 4: The section “2.3 Statistical Analysis” has been added, as suggested (line 92). The results are described in lines 173 and 187.
Comments 5: The sequences of DNA were analyzed by Sanger, but the Cryptosporidium species are not indicated in the result section. Please explain why or clarify in M&M section if all positive samples for both pathogens were sequenced.
Response 5: Sanger sequencing of all the amplicons was performed in order to confirm the identification, avoiding false positives due to non-specific reactions. This information has been added to the text (line 140).
Comments 6: Ethical and bioethical certifications were not included in the article. Please add this information.
Response 6: This information has been added to the text (line 302)
Minor suggestions:
Comments 7: Line 30, please replace “Cryptoporidiosis” by “Cryptosporidiosis”.
Response 7: The text has been corrected, as suggested (line 34)
Comments 8: Line 40, please add a reference after “kingdom Fungi”.
Response 8: The reference has been added, as suggested (line 44)
Comments 9: Line 45, write genera and species the first time they are mentioned (E. bieneusi and E. intestinalis).
Response 9: The text has been corrected, as suggested (line 49)
Comments 10: Line 85, please correct “1,3000”.
Response 10: The text has been corrected, as suggested (line 105)
Round 2
Reviewer 1 Report
Comments and Suggestions for Authors
That will be great
Reviewer 2 Report
Comments and Suggestions for Authors
The suggestions were included in the revised version. I think the article should be published.